# Access to General Practitioners during the COVID-19 pandemic in Portugal—A survey study of patient experiences in an urban setting

Mónica Granja [1,2]*, Luís Alves[1,2], Sofia Correia[1,2]

1 EPIUnit, Instituto de Saúde Pública, Universidade do Porto, Porto, Portugal, 2 Laboratório para a Investigação Integrativa e Translacional em Saúde Populacional, Universidade do Porto, Porto, Portugal

* monicagranja66@gmail.com

**Data Availability Statement:** All relevant data are within the paper and its Supporting Information files.

## Abstract

### Background

In 2020, Portugal had high levels of unmet health care needs. Primary Care was reported as the main source of unmet needs.

### Objectives

To describe face-to-face and remote access to GPs in Portugal during the COVID-19 pandemic. To discover patient experiences and attitudes to access to care. To identify determinants of access to care.

### Methods

A survey of a random sample of 4,286 adults registered in a group of Family Practices was conducted in 2021. Paper questionnaires were sent by post to patients who had no e-mail address registered with the practice. Patients with an e-mail address were sent a link to an online questionnaire. Outcomes were reported waiting times for face-to-face and remote contacts with GPs, dichotomized to ascertain compliance with standards. Associations between participant characteristics and outcome variables were tested using logistic regression.

### Results

Waiting times for face-to-face consultations with GPs during the pandemic often exceeded the maximum waiting times (MWT) set by the National Health Service. Remote contacts were mostly conducted within acceptable standards. Waiting times for speaking with the GP over the phone were rated as 'poor' by 40% and 27% reported requests for these calls as unmet. The odds of getting care over MWT increased for participants who reported poorer digital skills. Participants were less likely to get non-urgent consultations over MWT if they found it easy to use the online patient portal to book appointments (odds ratio 0.24; 99%

**Funding:** Matosinhos Local Health Unit (Portugal) funded this study, supporting the expenses of pre-stamped institutional envelopes for the paper questionnaire. Instituto de Saúde Pública, Universidade do Porto, supported the expenses of printing the paper questionnaires. The funders had no role in study design, data collection and analysis, decision to publish, or preparation of the manuscript.

**Competing interests:** The authors have declared that no competing interests exist.

confidence intervals 0.09–0.61), request prescriptions (0.18; 0.04–0.74) or insert personal data (0.18; 0.04–0.95).

## Conclusion

Patient reported access to GPs during the pandemic was uneven in Portugal. Obtaining non-urgent consultations and remote contacts over MWT affected mainly those patients with poor digital skills. Telephone access to GPs received the worse ratings. Access through traditional pathways must remain available, to prevent the widening of inequities.

## Introduction

Primary care-based systems may achieve better population health and are more equitable than those based on specialist, hospital or market-driven care [1, 2]. Easy access to General Practitioners (GPs) is essential in well-functioning Primary Care systems [3]. Accessibility has been defined as «the ease with which a person can obtain needed healthcare, from the practitioner of choice and within a timeframe appropriate to the urgency of the problem» [3]. Accessibility is a key feature of General Practice [3].

The Portuguese health system is based on Primary Care. The National Health Service (NHS) provides universal and mostly free coverage for a set of health care services. GPs are salaried and work in Family Practices owned by the state and clustered in 55 groups of Health Centres in Continental Portugal. Standards for maximum waiting times (MWT) for some services are set by the government [4]. Telephone and e-mail contacts with the GP have no established MWT [4]. The NHS online patient portal permits GP appointment booking, repeat prescription ordering, and personal health data recording.

Despite established maximum waiting times (MWT), compliance is uneven [5]. This may lead patients to seek more timely care in the private sector, resulting in high out-of-pocket health expenditures [6]. In 2015, out-of-pocket expenses for health care in Portugal were above the OECD average. This affected mostly poorer households [6]. Telephone contact between GPs and their patients is common in Portuguese Family Practices but the adoption of the use of electronic mail for contact between GPs and their patients has been restricted [7, 8]. In 2015, patient satisfaction with access to GPs, especially telephone access, was found to be lower than overall satisfaction with care [9]. Data are not publicly available in Portugal on whether MWT are deemed acceptable by service users, nor on the utilization of the NHS online patient portal.

During the COVID-19 pandemic access to care was impaired because healthcare workers were diverted to COVID-19 care [10]. Lockdowns, social distancing restrictions in healthcare facilities and mandatory isolation of staff also played a role [11, 12]. Mortality hit harder disadvantaged populations [13]. The shift from face-to-face to remote care affected most those with difficulties in the use of digital technologies, widening disparities [14]. Unmet care needs increased worldwide [15], affecting those with less income the most [16]. In 2020, Portugal ranked as the second lowest country on the OECD scale of unmet health care needs [6]. Primary Care was reported by people aged over 50 to be the main source of unmet needs in 2020 [17].

There is a need for updated data on accessibility to GPs in Portugal, particularly in the pandemic context. This study aimed to describe access to GPs in Portugal in the second year of the

COVID-19 pandemic, including face-to-face and remote access, to discover patient experiences and attitudes to access, and to identify determinants of access to care.

## Methods

### Design

We conducted a quantitative, descriptive, and cross-sectional survey-based research.

### Setting

The study setting was the Primary Care department of Matosinhos Local Health Unit, comprising the 14 Family Practices of an urban municipality. Matosinhos is located in the northern coastal region of Portugal. Its 172,557 inhabitants display an age distribution similar to the Portuguese population, with 13% of youths up to 14 and 23% elderly aged over 65 [15]. Illiteracy affects 4% of Matosinhos inhabitants (national average is 6%) while 25% attained higher education (national average is 20%) [18].

### Study sample

In May 2021 there were 151,081 persons aged over 18 registered in the Primary Care department of Matosinhos Local Health Unit. From the registration list, a random sample (n = 4,286) was drawn by the Information Technology department, using the Oracle random number generator.

In keeping with data protection regulations, the Information Technology department handled the recruitment mailings, so the researchers had no access to any personal data of the selected sample. Paper invitations were sent by post to selected persons who had no e-mail address registered with Matosinhos Local Health Unit (n = 2,674), along with a paper questionnaire and a prepaid envelope enclosed for returning the questionnaire, once completed. Selected persons with an e-mail address (n = 1,612) were sent an invitation by e-mail with a link to an online version of the questionnaire. Invitations and questionnaires were sent during May and June 2021. Two weeks after the first e-mail an e-mail reminder was sent to participants with an e-mail address. Responses were included in the analysis if they were received by post or e-mail by 31/08/2021.

Sample size was calculated for an expected proportion of 50% on most outcomes, with a confidence level of 95%, and a margin of error of 5%. Using a conservative approach based on known estimates, the number obtained (n = 384) was increased to 600 to guarantee power for inferential statistics and was further increased to cover an expected response rate of 14%. This considered an expected rate of outdated address information of 30% and a response rate of 20% among those who would receive an invitation to participate. This produced a target sample of 4,286 persons to invite.

### Instrument

A self-administered, structured, anonymous questionnaire was constructed following a literature review of studies on patient access to general practice [19–25]. GPs, epidemiologists, laypeople, and a statistician participated in an assessment of face validity and content validity of the questionnaire. A pilot study was conducted with a convenience sample of patients (n = 104) from four Family Practices, three of them outside the target area, including rural and urban settings and different types of Family Practice organization. Field observations and analysis of the responses led to changes in the final version of the questionnaire. Its length was

reduced and questions with many missing responses or ceiling/floor effects were reviewed or cancelled.

The final version of the questionnaire 'Accessibility to the General Practitioner' includes six sections (S1 Appendix). A first contact section comprises six items on first contact care and the reasons for each choice. There is a two-item section about Family Practice facilities and physical access. One section addresses face-to-face contacts, with seven items querying experienced waiting times and views on same times. Another section addresses remote contacts, with four items querying waiting times, views, and digital skills. Digital skills are assessed by the ratings given by participants to the usability of digital prescriptions and of the NHS patient portal. One section comprises five items about health services utilization and satisfaction, one of which was drawn from the EUROPEP patient satisfaction questionnaire [24]. The last section includes thirteen items on age, sex, education, employment status, marital status, household composition, use of telephone and internet and health status.

Participants' age and number of medicines regularly taken were collected as discrete variables. All other variables were collected either as ordinal, where applicable, or nominal.

## Outcomes

Outcomes were reported waiting times, regarding the previous 6 months. Time, collected as an ordinal variable, was dichotomized to ascertain compliance with standards. According to the maximum waiting times (MWT) for General Practice set by the government [4], requests for face-to-face appointments for acute illness were to be fulfilled on the same day, requests for appointments for non-acute reasons were to be met within 15 working days, requests for home visits were to be satisfied within 24 hours, and requests for remote prescription renewals and medical reports were to be completed within 72 hours. Three working days was the cut-off chosen by the researchers for responding to requests of phone calls and e-mails, for which standards are not set by the government. In dichotomization, unmet requests for remote contacts were considered as exceeding MWT or the defined cut-off.

## Analysis

Descriptive statistics were used to characterize study participants and their experiences and attitudes to access to health care. Comparisons between groups were tested with Student t-test for continuous variables and with $\chi 2$ and Fisher tests for categorical variables. An alpha level of 0.05 was used for these comparisons. Logistic regression analysed the relationship between participant characteristics and views and compliance with standards for waiting times, adjusting for sociodemographic and health characteristics. Missing data were deleted pairwise. Odds ratio with 99% confidence intervals were calculated for all models. IBM SPSS Statistics for Windows, Version 27.0 was used for the analysis.

The study protocol was approved by the ethics committee of Matosinhos Local Health Unit on 10/07/2020 (nr. 59/CE/JAS).

## Results

Out of the paper questionnaires sent, 252 were completed, returned, and included in the analysis for a response rate of 10%. The post office returned 73 forms as 'not delivered'. The online questionnaire returned 690 responses, 556 of which were included in the analysis for a response rate of 35%. Questionnaires without responses to questions about face-to-face consultations or remote contacts were excluded from the analysis. A total of 808 persons were included in the study, resulting in an overall response rate of 19% (Fig 1).

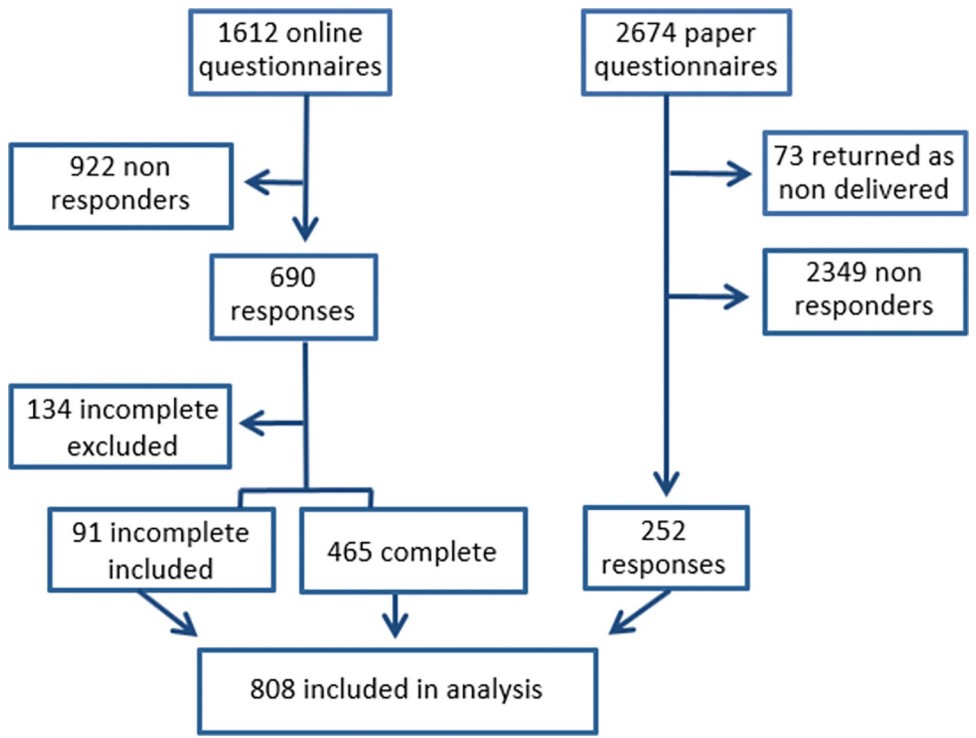

**Fig 1. Flowchart of participant recruitment to survey.**

The mean age of participants was 53.5 years (range 18–93), 58% were women, 60% had completed at least the 11th grade in high school and 51% were employed (Table 1).

Most participants were married or living in a common-law relationship. Over 80% reported regular use of internet. Most considered their health status to be good or very good and 40% were registered with the same GP for more than 10 years.

Compared to the original random sample selected, respondents were more often females and reported higher education levels. Participants responding to the paper questionnaire differed from the ones who answered online questionnaires. They were more often male, older, had lower education levels, were less often employed, were registered with a GP for longer periods and had poorer self-perception of their health status (Table 1).

## Accessibility to the General Practitioner

Access to face-to-face consultations with GPs was reported to exceed MWT by 54% for participants with urgent reasons, and by 56% with non-urgent reasons (Table 2). Waits over five working days for urgent consultations were reported by 16%, and 20% reported waiting over 3 months for non-urgent consultations. Of those requesting home visits, 83% reported getting them beyond the MWT. Remote contacts were most often reported to occur within the MWT (or three working days), even categorizing unmet requests for remote contacts as exceeding the MWT. The exceptions were requests for video consultations, attempted by 3% of participants and, most often, not accomplished. Requests for telephone calls with the GP were reported to occur within 1 working day by 36% of participants, and not to be met by 27%.

Waiting times for home visits, GP replies to e-mail contacts, and requests for remote review of test results were mostly rated as 'good' (Table 2). Waiting times for telephone calls and video consultations with the GP were mostly rated as 'poor'. Waiting times for office face-to-face

**Table 1. Participant characteristics and comparison with the original sample and the population in Portugal.**

| characteristics | all participants | online participants | paper participants | comparison online x paper | original sample | comparison participants x original sample |
|---|---|---|---|---|---|---|
| | n = 808 | n = 556 | n = 252 | p | n = 4286 | p |
| age (years) | | | | <0.001 | | 0.079 |
| mean (SD) | 53.5 (17.42) | 50.7 (16.41) | 59.0 (18.12) | | 52.2 (18.53) | |
| minimum—maximum | 18–93 | 18–92 | 18–93 | | 18–103 | |
| sex | | | | <0.001 | | 0.007 |
| female % | 57.5 | 62.1 | 48.4 | | 52.1 | |
| education (completed grade) % | | | | <0.001 | | <0.001 |
| $\leq$ 4th | 21.5 | 13.7 | 37.0 | | 29.6 | |
| 6th or 9th | 18.9 | 16.0 | 24.8 | | 25.9 | |
| 11th or 12th | 27.8 | 31.4 | 20.7 | | 24.1 | |
| university | 31.7 | 38.9 | 17.5 | | 20.3 | |
| employment status % | | | | <0.001 | | |
| employed | 50.9 | 58.5 | 35.6 | | | |
| retired | 30.7 | 23.9 | 44.5 | | | |
| unemployed | 7.9 | 9.0 | 5.5 | | | |
| others | 10.5 | 8.6 | 14.4 | | | |
| marital status | | | | 0.300 | | |
| married/common law % | 62.5 | 61.2 | 65.1 | | | |
| internet use% | | | | <0.001 | | |
| never | 11.1 | 5.3 | 23.8 | | | |
| seldom / some days | 5.2 | 4.1 | 7.6 | | | |
| often / every day | 83.7 | 90.6 | 68.6 | | | |
| perceived health status % | | | | | | |
| poor | 11.6 | 10.0 | 15.1 | 0.002 | | |
| fair | 29.9 | 26.7 | 36.6 | | | |
| good | 31.7 | 34.3 | 26.3 | | | |
| very good/excellent | 26.7 | 29.0 | 22.0 | | | |
| years registered with GP % | | | | | | |
| 0 -<1 | 12.4 | 12.2 | 12.8 | <0.001 | | |
| 1–4 | 21.5 | 25.0 | 13.6 | | | |
| 5–10 | 26.1 | 27.1 | 23.8 | | | |
| >10 | 39.9 | 35.6 | 49.8 | | | |

GP: General Practitioner

consultations and for remote contacts with the GP were mostly rated as 'fair'. What participants consider poor, fair or good varied across all types of contacts. Waits for home visits got mostly good ratings even when exceeding the MWT. Waiting times for a phone call to be returned got a large share of 'poor' ratings even when meeting the target of three working days.

## Digital skills

Most participants reported receiving prescriptions sent by text message (82%) or e-mail (61%). Over 90% of patients found it 'easy' or 'very easy' to use this service. A smaller proportion of patients used the NHS patient portal: 34% had used it to request repeat prescriptions, 30%

**Table 2. Distribution of waiting times to face-to-face consultations and remote contacts.**

| type of contact | % ever used | reported waiting times | | | | | % reporting unmet requests | participants' rating of waiting times | | |
|---|---|---|---|---|---|---|---|---|---|---|
| | | | | % | | | | (very) poor | fair | (very) good |
| | | | | | | | | % | | |
| | | working days | | | | | | | | |
| | | same day | 1–2 | 3–5 | >5 | | | | | |
| **urgent consultation** n = 763 | 63.7 | 45.7 | 26.1 | 12.1 | 16.0 | | - | 26.4 | 41.8 | 31.8 |
| | | weeks | | | | | | | | |
| | | 1 | 1–3 | 4–8 | 9–12 | > 12 | | | | |
| **non-urgent consultation** n = 752 | 79.0 | 16.5 | 27.1 | 26.4 | 10.4 | 19.5 | - | 29.0 | 49.1 | 21.9 |
| | | working days | | | | | | | | |
| | | 1 | 2–3 | 4–5 | >5 | | | | | |
| **home visit** n = 752 | 7.6 | 17.2 | 29.3 | 13.8 | 39.7 | | - | 16.7 | 34.8 | 48.5 |
| **prescriptions from front desk** n = 717 | 59.3 | 15.1 | 49.2 | 25.9 | 8.2 | | 1.6 | 8.8 | 48.3 | 43.0 |
| **prescriptions from patient portal** n = 693 | 14.3 | 16.2 | 43.4 | 17.2 | 12.1 | | 11.1 | 12.0 | 48.1 | 39.8 |
| **medical reports** n = 699 | 25.7 | 16.7 | 39.4 | 14.4 | 21.1 | | 8.3 | 18.2 | 44.8 | 37.0 |
| | | working days | | | | | | | | |
| | | 1 | 2–3 | 4–5 | >5 | | | | | |
| **investigations review** n = 703 | 38.7 | 18.0 | 33.5 | 19.1 | 19.1 | | 10.3 | 19.1 | 40.4 | 40.4 |
| **return phone call from GP** n = 719 | 38.6 | 36.3 | 24.8 | 7.2 | 4.7 | | 27.0 | 39.6 | 31.9 | 28.5 |
| **GP reply to e-mail** n = 696 | 66.1 | 35.9 | 33.7 | 10.9 | 8.9 | | 10.7 | 22.4 | 34.8 | 42.8 |
| **video consultation** n = 688 | 3.0 | 23.8 | 4.8 | 4.8 | 14.3 | | 52.4 | 50.0 | 40.0 | 10.0 |

GP: General Practitioner

green shading indicates compliance with maximum waiting times set by the Portuguese government [4]

blue shading indicates the target proposed by the researchers, since no government standards were set for these contacts.

used it to book appointments, and 18% for entering personal data. Among patients using the NHS portal, it was rated as 'difficult' by 45% for repeat prescriptions requests, by 21% for booking appointments, and by 35% for insertion of personal data (Table 3).

## Determinants of waiting times

Participants with a university degree were more likely to report getting urgent consultations over MWT (odds ratio 2.71; 99% confidence interval 1.06–6.90). The odds of getting non-

**Table 3. Participants' ratings of the usability of paper prescriptions and of digital services related to General Practice and comparison between online and paper participants.**

| ratings of usability | total n | never used | if ever used | | online x paper |
|---|---|---|---|---|---|
| | | | (very) difficult | (very) easy | |
| | | n (%) | n (%) | | p |
| **prescriptions** | | | | | |
| paper | 717 | 86 (12.0) | 41 (6.5) | 590 (93.5) | 0.108 |
| text message | 725 | 134 (18.5) | 50 (8.5) | 541 (91.5) | 0.245 |
| e-mail | 709 | 274 (38.6) | 28 (6.4) | 407 (93.6) | 0.013 |
| **patient portal** | | | | | |
| book appointments | 757 | 502 (66.3) | 113 (44.5) | 142 (55.7) | 0.351 |
| request prescriptions | 748 | 527 (70.5) | 47 (21.3) | 178 (78.7) | 0.054 |
| insert health data | 719 | 590 (82.1) | 42 (34.9) | 84 (65.1) | 0.382 |

urgent consultations over MWT decreased for participants who found it easy to use the NHS patient portal to book appointments (0.24; 0.09–0.61), to request prescriptions (0.18; 0.04–0.74) or insert personal data (0.18; 0.04–0.95). Getting repeat prescriptions over the MWT was less likely for those who were registered with the same GP for over 10 years (0.38; 0.15–0.94) and with a very good self-perceived health status (0.24, 0.08–0.75), and for those those who found it easy to use the NHS portal to book appointments (0.13; 0.03–0.50) and to request prescriptions (0.11; 0.03–0.47). Being registered with the same GP for over 10 years also decreased waits over 3 working days for contacts with the GP by telephone (0.25; 0.08–0.79) and e-mail (0.30; 0.13–0.72). Waits over 3 working days for telephone contacts with the GP were less likley for participants with a very good self-perceived health status (0.24, 0.06-.089) and for those who found it easy to get prescriptions by e-mail (0.07; 0.01–0.71) and on the NHS portal (0.02; 0.00–0.27) (Table 4).

## Discussion

### Main findings

In this survey study, aimed to assess accessibility to GPs in Portugal during the second year of the COVID-19 pandemic, most participants reported waiting times for face-to-face consultations exceeding MWT, while remote contacts complied with MWT. This may be related to a shift from face-to-face consultations to remote care caused by the pandemic [26]. However, non-compliance with MWT for face-to-face care was also found before the pandemic [27]. Pre-pandemic explanations may be increasing list size, ageing and medical complexity of the population, medicalization, and escalating paperwork. Aggravating factors during the pandemic may be related to extra tasks assigned to GPs, such as COVID-19 follow-up calls and work in acute respiratory disease clinics and vaccination centres.

Waiting times for a telephone call to be returned by the GP were mostly rated as poor by participants, even when within the standards. The opposite was found for home visits. This suggests that the adopted targets may be inappropriate. Moreover, the waiting times that participants considered poor, fair, or good varied substantially. This may mean that patients value other aspects over rapid access. Timeliness may be traded-off against continuity of care, good communication with the doctor, or preference for a convenient time of day [28, 29].

Though most participants reported regular use of the internet, most had not used the NHS patient portal, and many users found it difficult to navigate. Poor ratings of the usability of NHS digital services were associated with reported waits over time standards, but low educational levels were not. Digital skills are essential for digital communication [14] and digital exclusion is a concern among patients [11]. The shift from face-to-face to remote care caused by the pandemic has exposed digital exclusion as an emergent determinant of health inequalities [30]. Other factors associated with decreased odds of getting remote care over MWT (or 3 working days when MWT are not set) were being registered with the same GP for over 10 years and a very good self-perceived health status. Shorter waiting times may be one of the mechanisms underlying the reported benefits of provider continuity [31]. Also, longer doctor-patient relationships are more likely to occur with older GPs. A doctor survey conducted simultaneously with this patient survey found that older GPs reported shorter waiting times for their services [32]. As for self-perceived health status, healthier patients, having less care needs, are less likely to experience delays at some stage.

We found a large proportion of unmet requests for telephone access to the GP and dissatisfaction with waiting times for telephone contacts with the GP. The expansion of the Portuguese medical telephone triage service, SNS24, was inadequate in meeting the pandemic surge

**Table 4. Adjusted odds ratio of reporting response over time standards on main outcomes.**

| | | urgent consultation | non-urgent consultation | prescription at front desk | telephone call with GP | e-mail with GP |
|---|---|---|---|---|---|---|
| | | over maximum waiting time | | | over 3 working days | |
| | | OR adjusted§ [99% CI] | | | | |
| **sex** | female | | | | | |
| | male | 1.06 [0.61–1.86] | 0.71 [0.43–1.16] | 1.10 [0.60–2.02] | 0.74 [0.34–1.60] | 1.30 [0.70–2.42] |
| **age** | <40 | | | | | |
| | 40–54 | 1.27 [0.60–2.69] | 1.18 [0.61–2.31] | 0.59 [0.21–1.71] | 0.81 [0.29–2.29] | 0.93 [0.44–2.00] |
| | 55–64 | 2.09 [0.87–5.03] | 1.35 [0.61–3.00] | 0.93 [0.31–2.81] | 0.68 [0.20–2.29] | 0.81 [0.32–2.07] |
| | 65–74 | 1.38 [0.50–3.75] | 0.90 [0.38–2.11] | 0.54 [0.16–1.75] | 0.53 [0.13–2.07] | 0.88 [0.31–2.48] |
| | ≥ 75 | 1.67 [0.52–5.40] | 1.08 [0.40–2.92] | 0.76 [0.22–2.58] | 0.88 [0.22–3.47] | 0.50 [0.13–1.89] |
| **marital status** | married | | | | | |
| | unmarried | 0.69 [0.39–1.23] | 1.46 [0.88–2.44] | 0.68 [0.35–1.30] | 1.19 [0.53–2.68] | 1.32 [0.71–2.43] |
| **education** | ≤ 4th | | | | | |
| | 6th or 9th | 1.08 [0.44–2.65] | 1.02 [0.46–2.30] | 0.93 [0.39–2.22] | 0.61 [0.20–1.89] | 0.55 [0.18–1.67] |
| | 11th or 12th | 1.68 [0.69–4.11] | 1.12 [0.50–2.52] | 0.90 [0.36–2.24] | 0.74 [0.23–2.34] | 0.66 [0.22–2.00] |
| | university | **2.71 \* [1.06–6.90]** | 1.43 [0.65–3.18] | 1.33 [0.53–3.36] | 1.28 [0.39–4.26] | 0.99 [0.34–2.84] |
| **years with same GP** | 0-<1 | | | | | |
| | 1–4 | 1.23 [0.47–3.25] | 0.49 [0.20–1.16] | 0.37 [0.13–1.07] | **0.21\* [0.06–0.75]** | 0.63 [0.25–1.60] |
| | 5–10 | 0.90 [0.36–2.28] | 0.43 [0.18–1.02] | 0.58 [0.22–1.52] | 0.29 [0.08–1.04] | **0.29\* [0.11–0.73]** |
| | >10 | 0.61 [0.26–1.43] | 0.48 [0.21–1.07] | **0.38\* [0.15–0.94]** | **0.25\* [0.08–0.79]** | **0.30\* [0.13–0.72]** |
| **self-perceived health status** | poor | | | | | |
| | fair | 0.54 [0.23–1.30] | 1.01 [0.43–2.38] | 0.56 [0.25–1.29] | 0.63 [0.22–1.87] | 0.63 [0.23–1.76] |
| | good | 0.41 [0.16–1.03] | 0.87 [0.36–2.10] | **0.39\* [0.15–0.97]** | 0.45 [0.13–1.50] | 0.44 [0.16–1.26] |
| | very good | 0.65 [0.23–1.81] | 0.65 [0.26–1.64] | **0.24\* [0.08–0.75]** | **0.24\* [0.06–0.89]** | 0.38 [0.13–1.16] |
| **prescriptions by text message** | difficult | | | | | |
| | easy | 1.57 [0.54–4.61] | 0.50 [0.16–1.55] | 0.55 [0.19–1.56] | 0.74 [0.21–2.64] | 0.55 [0.18–1.71] |
| **prescriptions by e-mail** | difficult | | | | | |
| | easy | 0.30 [0.05–1.79] | 0.21 [0.04–1.23] | 0.24 [0.05–1.25] | **0.07\* [0.01–0.71]** | **0.11\* [0.02–0.52]** |
| **book appointment on patient portal** | difficult | | | | | |
| | easy | 0.83 [0.31–2.25] | **0.24\* [0.09–0.61]** | **0.13\* [0.03–0.50]** | 0.26 [0.06–1.06] | 0.47 [0.17–1.29] |
| **request prescriptions on patient portal** | difficult | | | | | |
| | easy | 0.23 [0.05–1.17] | **0.18\* [0.04–0.74]** | **0.11\* [0.03–0.47]** | **0.02\* [0.00–0.27]** | 0.41 [0.11–1.52] |
| **insert data on patient portal** | difficult | | | | | |
| | easy | 0.34 [0.07–1.69] | **0.18\* [0.04–0.95]** | 0.11 [0.01–1.24] | 0.03 [0.00–2.64] | 0.31 [0.06–1.28] |

§Odds Ratios were adjusted for sex, age, marital status, education, years registered with the same General Practitioner and self-perceived health status; *statistically significant associations; GP: General Practitioner; OR: Odds Ratio; CI: confidence interval.

in demand. Telephone access to Portuguese Family Practices, a pre-pandemic weakness, remained poor [33]. Knowing this, the national Public Health Service directed GPs to make daily calls to infected patients [34]. These directions were in effect up until January 2022. They resulted in even busier telephone lines and an increased workload for GPs, missing an opportunity for promoting the growth of telemedicine [35].

## Strengths and limitations

To our knowledge, this is the first study in Portugal exploring face-to-face and remote access to GPs during the pandemic. We were able to recruit a good size random sample of people registered in PC, irrespective of user status. The use of a non-validated questionnaire is a limitation. The absence of a validated questionnaire addressing both face-to-face and remote access applicable to the Portuguese context created the need to construct a new questionnaire. The instrument was tested in a pilot study that informed amendments to the questionnaire. The response rate of 19% is another limitation. A reminder message was sent to online participants, reducing non-response. Budget constraints prevented mailing of reminders to paper participants, but this format increased the recruitment of male and less-educated participants, characteristics associated with selective non-response in our sample. Controlling for these characteristics, and for sex and self-perceived health status (also known to associate with selective non-response), in multivariate analysis, minimized the effects of non-response bias. The study setting (Matosinhos) was a single regional group of Health Centres (among 55 such groups nationwide), limiting generalizability. However, the original sample was similar to the Portuguese population regarding sex, education, and internet access, as this municipality displays demographic similarities with the Portuguese population [18]. The risk of recall bias, as information was requested regarding behaviour in the previous 6 months, and of under-reporting bias, must be acknowledged.

## Implications for research and practice

This study shows a digital divide impairing equity in access to healthcare in Portugal. Access to traditional, non-digital pathways, such as face-to-face consultations and telephone contacts, must remain available. Barriers to telephone access to Family Practices must be overcome. Digital communication media for access to care need platforms that are more user-friendly and secure. At a macro-level, policy makers must address digital illiteracy in the Portuguese population.

The review and future definition of MWT may be informed by our participants' assessment of waiting times. The incomplete overlap between waiting time and ratings given to waiting time could be further explored using qualitative methodologies.

The findings of this study may inform the inclusion of questions about remote access in the EUROPEP questionnaire. A validated questionnaire on accessibility in General Practice would allow for repeat measures and quality improvement.

Provider determinants of long waiting times for consultations with GPs need to be addressed to ensure access and real universal coverage.

## Conclusion

Patient reported access to GPs during the pandemic was uneven in Portugal. Long waiting times for face-to-face consultations and remote contacts affected most those patients with poorer skills in the use of digital media. Waiting times for telephone access to GPs received the worse ratings. Access through traditional pathways must remain available in Family Practices, to prevent widening inequities in access to GPs.

## Supporting information

**S1 Appendix. Questionnaire.**
(PDF)

**S1 Table. Odds ratio of reporting response over maximum waiting times when requesting a non-urgent consultation.** OR: Odds Ratio; IC: confidence intervals; SD: sociodemographics (sex, age, marital status and education); GP: General Practitioner. *health variables; years registered with the same General Practitioner; self-perceived health status. bold: statistically significant.
(PDF)

**S2 Table. Odds ratio of reporting response over maximum waiting times when requesting an urgent consultation.** OR: Odds Ratio; IC: confidence intervals; SD: sociodemographics (sex, age, marital status and education); GP: General Practitioner. *health variables; years registered with the same General Practitioner; self-perceived health status. bold: statistically significant.
(PDF)

**S3 Table. Odds ratio of reporting response over maximum waiting times when requesting a repeat prescription at the front desk.** OR: Odds Ratio; IC: confidence intervals; SD: sociodemographics (sex, age, marital status and education); GP: General Practitioner. *health variables; years registered with the same General Practitioner; self-perceived health status. bold: statistically significant.
(PDF)

**S4 Table. Odds ratio of reporting response over three working days when contacting the GP by telephone.** OR: Odds Ratio; IC: confidence intervals; SD: sociodemographics (sex, age, marital status and education); GP: General Practitioner *health variables; years registered with the same General Practitioner; self-perceived health status. bold: statistically significant.
(PDF)

**S5 Table. Odds ratio of reporting response over three working days when contacting the GP by e-mail.** OR: Odds Ratio; IC: confidence intervals; SD: sociodemographics (sex, age, marital status and education); GP: General Practitioner *health variables; years registered with the same General Practitioner; self-perceived health status. bold: statistically significant.
(PDF)

**S6 Table. Participant's comparison of EUROPEP accessibility dimensions between pandemic and pre-pandemic times.** GP: General Practitioner.
(PDF)

**S1 File. Dataset.**
(SAV)

## Acknowledgments

The authors wish to thank:

All the people who participated in the pilot and in the main study.

Matosinhos Local Health Unit for supporting the random sample process.

The GPs and lay people who participated in the face and content validation of the questionnaire.

The principals of the four Family Practices where the pilot study was run.

John Yaphe for the English editing and the critical review of the manuscript.

## Author Contributions

**Conceptualization:** Mónica Granja, Luís Alves, Sofia Correia.

**Data curation:** Mónica Granja.

**Formal analysis:** Mónica Granja, Luís Alves, Sofia Correia.

**Funding acquisition:** Mónica Granja, Sofia Correia.

**Investigation:** Mónica Granja, Luís Alves, Sofia Correia.

**Methodology:** Mónica Granja, Luís Alves, Sofia Correia.

**Project administration:** Mónica Granja, Luís Alves, Sofia Correia.

**Resources:** Mónica Granja, Luís Alves, Sofia Correia.

**Supervision:** Luís Alves, Sofia Correia.

**Validation:** Luís Alves, Sofia Correia.

**Writing – original draft:** Mónica Granja, Sofia Correia.

**Writing – review & editing:** Mónica Granja, Luís Alves, Sofia Correia.

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
