## [Decision Letter · Decision Letter 0]

17 Feb 2023

PONE-D-23-02591

Access to General Practitioners during the COVID-19 pandemic in Portugal

PLOS ONE

Dear Dr. Granja,

Thank you for submitting your manuscript to PLOS ONE. After careful consideration, we have decided that your manuscript does not meet our criteria for publication and must therefore be rejected.

I am sorry that we cannot be more positive on this occasion, but hope that you appreciate the reasons for this decision.

Kind regards,

Patricia Rezende do Prado

Academic Editor

PLOS ONE

Additional Editor Comments :

Dear authors,

Plos one uses unpublished publications as a criterion. In this way, we observe that there is already a similar paper published with the same theme:

Access to General Practitioners during the second year of the COVID-19 pandemic in Portugal: a nationwide survey of doctors. Granja M, et al. BMC Prim Care. 2023. PMID: 36782109.

We hope that this decision will not prevent the new submission in the future papers.

Kind regards

Reviewers' comments:

Reviewer's Responses to Questions

**Comments to the Author**

1. Is the manuscript technically sound, and do the data support the conclusions?

Reviewer #1: Partly

Reviewer #2: Yes

Reviewer #3: Partly

2. Has the statistical analysis been performed appropriately and rigorously? 

Reviewer #1: Yes

Reviewer #2: Yes

Reviewer #3: No

3. Have the authors made all data underlying the findings in their manuscript fully available?

Reviewer #1: Yes

Reviewer #2: Yes

Reviewer #3: Yes

4. Is the manuscript presented in an intelligible fashion and written in standard English?

Reviewer #1: Yes

Reviewer #2: Yes

Reviewer #3: No

5. Review Comments to the Author

Reviewer #1: Access to General Practitioners during the COVID-19 pandemic in Portugal

Title in page 1 is different than that in page 8

Abstract

• Background is not reflecting any thing related to how access to general practitioners during the COVID-19 pandemic affected

• Aim: more than one in the same statement it can be divided into 3 objectives

• Methods section is missing some important data like data related collection process. Mention of unnecessary data like the details about the statistics used.

• Recommendation needs to be added and also further research needed.

Introduction section

• Needs to be more comprehensive and address all key words; attitudes to access, and to

• determinants of access to care in term of the significance to access to GPs services

• Table 1 content; the essential related points can be summarized and directly stated

• There is a need to use other studies/ reports for further support of the problem and significance of the current research.

Materials and Methods:

Design: not mentioned (A quantitative, descriptive research design)

Setting:

Needs more clear description.

Study sample

• Needs to be mentioned clearly under the related title ( Study sample title is not mentioned)

• The major characteristics of the population studied not well described

• the method of selecting the sample not clearly described

• exclusion criteria are not mentioned

Instruments

• This section needs more organization

• Mixed details which are not related to the instrument section ( mixing between methods and the tool

• Specification of tool title, parts, number of items below each part not stated.

• Scoring of the tool parts need more clarification

• Tool validity and reliability; Reliability of the tool is not mentioned

Data collection

Needs specification of the duration of data collection

Description of the data collection process needs to be more organized.

Result section

Line 139-147 contains data need to be in the sample and methods section

Tables and figures

• Table2 headings too long

• Table 2 includes comparison which I question its significance in the table.

• Line 164-169 contains data not need

• Table top raw has the total though each raw has its own total

• Table 3:

• This table includes too much details and it is not clear

• The rating not clear for what. if it is to the waiting time then very poor to very good is not suitable for description

• Line 190- 191 I question what is the significance of the comparison ?

• Line 190- 191 Can be moved to the discussion section

• comment on the Table 5 is not clear.

Discussion

Introductory statement and the aim need to be added

Description in the discussion section is not clear

Line 235-37 not related.

241 rating is not clear rating of what?

The details showing the comparability of other researches to this study not elaborated

Implications for research and practice

needs to be in a more summarized way showing directly the significance of study results.

Conclusion and recommendations:

Conclusion is not covering patient experiences and attitudes to access, and determinants of access to care

There is no recommendations made for practice and further research

References

13 out of 30 from 2016 and before

APA reference style:

Author Last name, First initial. Middle initial. (Year Published). Title of article.

Title of Periodical, Volume(Issue), page range. https://doi.org/xxxx or URL

Ex: Burnell, K. J., Coleman, P. G., & Hunt, N. (2010). Coping with traumatic memories:

Second World War veterans’ experiences of social support in relation to the narrative coherence

of war memories. Ageing and Society, 30(1), 57-78. https://doi.org/10.1017/S0144686X0999016X

Reviewer #2: This is an interesting study and the authors have collected a unique dataset using cutting edge methodology. The paper is generally well written and structured. thank you very much for submitting your paper

Reviewer #3: The title of the manuscript is good. However, it does need an edition by a native English speaker or Journal English language services. In addition on the result part the table should be edited accordingly. There are many significant numeric contradictions of the study participants on table two. On table five, please put starlike symbol (*) on the significant variables.

6. PLOS authors have the option to publish the peer review history of their article (what does this mean?). If published, this will include your full peer review and any attached files.

Reviewer #1: No

Reviewer #2: No

Reviewer #3: **Yes: **Bahredin Abdella Abdlsemed, Lecturer in Department of Medicine, College of Medicine and Health Science, Werabe University, Worabe, Ethiopia

- - - - -

---

## [Author Response · Author response to Decision Letter 0]

2 Apr 2023

Dr Emily Chenette, PhD

Editor-in-Chief

Plos One

March 6, 2023

Dear Dr. Chenette,

We would like to thank the Editor for this opportunity to review our manuscript ‘Access to General Practitioners during the COVID-19 pandemic in Portugal - A survey study of patient experiences in an urban setting’ based on your reviewers’ comments. Especially, we would like to thank the reviewers for their careful reading of the paper and for their thoughtful comments. We believe this has helped us to improve the quality of the paper. 

Please find below our point-by-point response. 

We hope our paper can now be accepted for publication.

Sincerely,

Mónica Granja, MD

Luís Alves, PhD

Sofia Correia, PhD

EPIUnit - Instituto de Saúde Pública, Universidade do Porto, Rua das Taipas, n° 135, 4050-600 Porto, Portugal

Laboratório para a Investigação Integrativa e Translacional em Saúde Populacional (ITR), Universidade do Porto, Rua das Taipas, n° 135, 4050-600 Porto, Portugal

 

[Editor and reviewers’ comments are transcribed in bold typeface]

[Line numbers are those in the clean version of the manuscript]

Editor Comments to the Author

Plos one uses unpublished publications as a criterion. In this way, we observe that there is already a similar paper published with the same theme:

Access to General Practitioners during the second year of the COVID-19 pandemic in Portugal: a nationwide survey of doctors. Granja M, et al. BMC Prim Care. 2023. PMID: 36782109.

We believe there is a misunderstanding regarding this paper we have just published in BMC Primary Care because it is an entirely different study. 

Comparing both studies (the one just published and the one we have submitted to Plos One), each paper studied a totally different population (a national census of family doctors/GPs working in Portugal versus a random sample of users of a local health service, respectively), using different surveys, with different questions and perspectives, seeking different determinants of accessibility and performing different analysis. 

Review Comments to the Author

Reviewer #1: 

Access to General Practitioners during the COVID-19 pandemic in Portugal Title in page 1 is different than that in page 8

This was corrected. On page 1 there was a full title and a short title which were merged into the full title on page 1: ‘Access to General Practitioners during the COVID-19 pandemic in Portugal - A survey study of patient experiences in an urban setting’. The short title was eliminated. 

Abstract

• Background is not reflecting any thing related to how access to general practitioners during the COVID-19 pandemic affected

This was reviewed as requested. The original content was replaced by a statement about access during the pandemic.

• Aim: more than one in the same statement it can be divided into 3 objectives

Objectives in the abstract were rephrased into 3 sentences.

• Methods section is missing some important data like data related collection process. Mention of unnecessary data like the details about the statistics used.

Data were added regarding the mode of questionnaire administration. Some details about the analysis were cut.

• Recommendation needs to be added and also further research needed.

The main recommendation was added to the abstract as requested.

Introduction section

• Needs to be more comprehensive and address all key words; attitudes to access, and to determinants of access to care in term of the significance to access to GPs services

The introduction was re-written to address all key-words more comprehensively, as well as attitudes to access in Portugal, determinants of access. 

• Table 1 content; the essential related points can be summarized and directly stated

Table 1 was removed and its content was summarized in lines 65-72. All tables were renumbered.

• There is a need to use other studies/ reports for further support of the problem and significance of the current research.

Statements and references were added framing with more detail the issue of access to care, in particular Primary Care in Portugal, during the pandemic (paragraph 83-91).

Materials and Methods:

Design: not mentioned (A quantitative, descriptive research design)

This information was added to the Methods section (line 99).

Setting:

Needs more clear description.

The study setting was further detailed in lines 101-106.

Study sample

• Needs to be mentioned clearly under the related title ( Study sample title is not mentioned)

A ‘study sample’ level 2 sub-heading was added. For coherence, five other sub-headings were also added in the Methods section: Design, setting, instrument, outcomes and analysis.

• The major characteristics of the population studied not well described

Population characteristics were further detailed in the description of the study setting (lines 101-106).

• the method of selecting the sample not clearly described

A random number generator (Oracle) was used, this information was added (lines 108-111).

• exclusion criteria are not mentioned

No exclusion criteria were applied. 

Instruments

• This section needs more organization

An ‘Instrument’ subheading was added, and its content was reorganized.

• Mixed details which are not related to the instrument section (mixing between methods and the tool

Tool-related content was placed on the ‘Instrument’ subheading.

• Specification of tool title, parts, number of items below each part not stated.

Tool details were added as requested (lines 138-149).

• Scoring of the tool parts need more clarification

Formal scoring was not performed. Answer options were nominal and ordinal (lines 150-152). 

• Tool validity and reliability; Reliability of the tool is not mentioned

The questionnaire was not validated. This issue was addressed in ‘strengths and limitations’ section in the discussion, but some of this content was moved to the methods section, as suggested (lines 137-139). Actions on the results of the pilot study remain summarized in the discussion. 

We can also include the following details if the reviewers feel that they are necessary. The pilot study aimed to assess problematic questions or suggested answers, including layout and language issues, and the time required to complete the questionnaire. Though it included some quantitative assessment, it was not intended as a formal validation of the questionnaire. There were 104 responses (23 online and 81 paper). Field observations and analysis of the responses led to changes in the final version of the questionnaire. The pilot version of the questionnaire was too long. Headers were streamlined and several questions were eliminated. Questions with many missing answers were eliminated. Conditional questions did not work well in the paper version, so this format was kept to a minimum (in the first five questions). Options that were never chosen in otherwise well-functioning questions were removed. Options with ceiling/floor effects were either eliminated or broken down into more granular data, like shorter time intervals. Questions with longer multidimensional arrays did not work well in the paper version and were revised to shorter formats.

Data collection

Needs specification of the duration of data collection

The sentences regarding duration of data collection (formerly in lines 89 and 94-95) were brought closer together for clarity (lines 119-122). 

Description of the data collection process needs to be more organized.

Data collection process was re-written and reorganized (112-118).

Result section

Line 139-147 contains data need to be in the sample and methods section

Data on the number of paper and online questionnaires were moved to the methods section (lines 115 and 117).

Tables and figures

• Table2 headings too long

Table (renumbered as Table 1) heading was shortened.

• Table 2 includes comparison which I question its significance in the table.

The comparison with the population in Portugal was removed from this Table (renumbered as Table 1) and from the text below.

We believe we should keep the comparison between online and paper participants. It shows that the paper questionnaire, however expensive and yielding a lower response rate (compared to the online questionnaire), played a role in reaching participants who, usually, are underrepresented in survey research, such as the elderly, the male sex, and the lower educated. For the same reasons, we would also keep the comparison between the study sample and the original sample since it can help on the understanding of the non-response bias. In our sample, male sex and lower education levels were confirmed as associated with selective nonresponse (discussion lines 315-317).

• Line 164-169 contains data not need

The comparison with population in Portugal was removed from the text.

• Table top raw has the total though each raw has its own total

Total n on top row were kept, while n on each row (pertaining to valid n in each item) were deleted.

• Table 3:

• This table includes too much details and it is not clear

This table (renumbered as Table 2) was revised. Column headings were rephrased. The last column and some details were deleted. 

• The rating not clear for what. if it is to the waiting time then very poor to very good is not suitable for description

Ratings are given by participants to the waiting times they report. This was rephrased on the table, for clarity. In the questionnaire the items read: ‘How do you rate the waiting time for…’ and the answer options were ‘never tried’, ‘very poor’, ‘poor’, ‘fair’, ‘good’, and ‘very good’. This translation from the Portuguese questionnaire was reviewed by a native English speaker who is a Family Physician and an experienced researcher. A scale from poor to excellent is usually accepted for evaluative questions (Fowler FJ. Commonly used measurement dimensions. In: Improving survey questions – Design and evaluation. California: Sage Publications Inc.; 1995. pp. 156-165).

• Line 190- 191 I question what is the significance of the comparison?

We agree the clinical significance of the comparison is questionable and the comparison was removed both from the text and the table (renumbered Table 2).

• Line 190- 191 Can be moved to the discussion section

As above, line was deleted.

• comment on the Table 5 is not clear.

The footnote on this table (renumbered as Table 4) was rephrased.

Discussion

Introductory statement and the aim need to be added

An introductory statement was added as requested (lines 260-261).

Description in the discussion section is not clear

The discussion, and the rest of the manuscript, were edited by a native English speaker who is also a GP and an experienced researcher.

Line 235-37 not related.

We acknowledge that the focus of our study was access during the pandemic but we believe it cannot be overlooked that noncompliance with standards for waiting times was already an issue before the pandemic. If possible, we would like to keep these 2 sentences (now lines 265-267).

241 rating is not clear rating of what?

The sentence was rephrased (line 270).

The details showing the comparability of other researches to this study not elaborated

A reference was added to a study we have published recently drawn from a doctor survey on accessibility (lines 288-290). To our knowledge, this is the first study in Portugal assessing face-to-face and remote access to GPs during the pandemic. We have not found other comparable studies conducted abroad.

Implications for research and practice needs to be in a more summarized way showing directly the significance of study results.

This sub-section was re-written, summarized and related to study results.

Conclusion and recommendations:

Conclusion is not covering patient experiences and attitudes to access, and determinants of access to care

The conclusion was rephrased to clarify this point.

There is no recommendations made for practice and further research

Recommendations were added in this section.

References

13 out of 30 from 2016 and before

More recent references were added (ref nr. 10, 11, 15, 16, 17, 32). The reference manager used did not allow for tracking these changes. 

APA reference style:

Author Last name, First initial. Middle initial. (Year Published). Title of article.

Title of Periodical, Volume(Issue), page range. https://doi.org/xxxx or URL

Ex: Burnell, K. J., Coleman, P. G., & Hunt, N. (2010). Coping with traumatic memories:

Second World War veterans’ experiences of social support in relation to the narrative coherence

of war memories. Ageing and Society, 30(1), 57-78. https://doi.org/10.1017/S0144686X0999016X

We used Vancouver style (and not APA), as recommended by Plos One. 

Reviewer #2:

This is an interesting study and the authors have collected a unique dataset using cutting edge methodology. The paper is generally well written and structured. thank you very much for submitting your paper

Reviewer #3: 

The title of the manuscript is good. However, it does need an edition by a native English speaker or Journal English language services. 

The manuscript was fully edited by a family physician/GP who is a native English speaker and an experienced researcher. 

In addition on the result part the table should be edited accordingly. 

The tables were edited, as above.

There are many significant numeric contradictions of the study participants on table two. 

Total n on top row were kept, while n on each row (pertaining to valid n in each item) were deleted.

On table five, please put starlike symbol (*) on the significant variables.

On table (now renumbered) 4 an asterisk was added on significant variables, as requested.

---

## [Editor Report · Decision Letter 1]

4 May 2023

Access to General Practitioners during the COVID-19 pandemic in Portugal - A survey study of patient experiences in an urban setting

PONE-D-23-02591R1

Dear Dr. Granja,

We’re pleased to inform you that your manuscript has been judged scientifically suitable for publication and will be formally accepted for publication once it meets all outstanding technical requirements.

Kind regards,

Angélica Baptista Silva, Ph.Sc.

Academic Editor

PLOS ONE

Journal Requirements:

Additional Editor Comments (optional):

Dear Authors

Although there is an article about the same research, the contents are quite different, the first article presents data related to GPs, and the other presents data related to patients' perceptions. I consider that the article is adequate, as the gaps presented by the reviewers were satisfactorily resolved. The topic is crucial for advancing public health, and we must research and publish more about the patient's perceptions.
---

## [Editor Report · Acceptance letter]

15 May 2023

PONE-D-23-02591R1 

Access to General Practitioners during the COVID-19 pandemic in Portugal - A survey study of patient experiences in an urban setting 

Dear Dr. Granja:

I'm pleased to inform you that your manuscript has been deemed suitable for publication in PLOS ONE. Congratulations! Your manuscript is now with our production department. 

Kind regards, 

on behalf of

Dr. Angélica Baptista Silva 

Academic Editor

PLOS ONE